# Enantiomers and Their Resolution

Rodrigo Santos [1,2], Karen V. Pontes [1,*] and Idelfonso B. R. Nogueira [2,*]

1 Programa de Pós-Graduação em Engenharia Industrial (Industrial Engineering Program),
Escola Politécnica (Polytechnique Institut), Universidade Federal da Bahia, Salvador 40210-630, Brazil;
vergne_6@hotmail.com
2 Laboratory of Separation and Reaction Engineering, Associate Laboratory LSRE-LCM,
Department of Chemical Engineering, Faculty of Engineering, University of Porto, 4200-465 Porto, Portugal
* Correspondence: karenpontes@ufba.br (K.V.P.); idelfonso@fe.up.pt (I.B.R.N.)

**Definition:** Enantiomers share the same chemical formula but have different chemical structures, i.e., type of isomers. Enantiomers are present in several drugs, perfumes, food, and are a fundamental part of biomolecules. This subject is highly important for pharmaceutical companies. Enantiomeric drugs present different actuation in the human body; depending on the compound, one might combat the symptom, whereas its pair might cause damage. The separation of pairs of enantiomers requires a chiral environment that provokes a structural imbalance that conventional methods cannot provide. Enantioresolution is one of the most promissory studies that benefit several areas, such as pharmaceutical, biotechnology, food industry, and fine chemistry. Its resolution is of great importance, therefore, its main mechanisms of resolution will be explained herein.

**Keywords:** enantiomer; chirality; nomenclature; market; enantioresolution

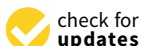



## 1. Introduction

Isomers are different compounds that share the same chemical formula but have different chemical structures. They are classified into structural and stereoisomers. Structural isomers might be subclassified into function, chain, position, metamerism and tautomerism, as exemplified in Table 1, the latter only existing in equilibrium. Stereoisomers are subclassified into diastereomers (diastereoisomers) and enantiomers (optical isomers). Isomers have different physicochemical properties such as their melting point, boiling point, solubility, and density. Enantiomers, unlike other isomers, share all these properties, but optical activity. Enantiomers are analogous to a pair of hands, an enantiomer is a specular image of the other.

Enantiomerism is present in several drugs, perfumes, food, and even in our own body. The response of human organisms to certain enantiomeric compounds might change dramatically depending on the enantiomer: one enantiomer in a drug might treat the disease whereas the other might cause harmful side effect. Thalidomide's side effect is, perhaps, the most remarkable and unfortunate known case in the literature. It is a drug that was used by pregnant women in the 1960's in the United Kingdom in order to combat morning sickness. However, the drug caused several birth defects [1]. This drug was sold in an equimolar mixture of enantiomer in which only one of them combats the symptom (eutomer), and the other caused side effects (deutomer) [2].

Large pharmaceutical companies aim to synthesize drugs that are quickly able to combat symptoms or illnesses with the least number of side effects by removing the deutomer. For this reason, enantioresolution is a crucial issue in the pharmaceutical industry. Distillation, decantation, and filtration are common methods of separation for compounds based on boiling point, density, and size, respectively. The resolution of compounds with different physicochemical properties might be simpler in comparison to enantioresolution. The separation of pairs of enantiomers, though, is not straightforward, and conventional

methods are nearly ineffective. Crystallization, membrane, or chromatography are examples of processes used to separate enantiomers in a chiral environment, molecules that tend to bind to an enantiomer instead of the other [3].

**Table 1.** Types of isomers and their structural formula [3].

| Isomerism | | Chemical Formula | Structural Formula | |
|---|---|---|---|---|
| Structural | Function | $C_3H_6O$ | Propanal | Propanone |
| | Chain | $C_4H_{10}$ | n-Butane | Isobutane |
| | Position | $C_5H_{10}O$ | 2-Pentanone | 3-Pentanone |
| | Metamerism | $C_4H_{10}O$ | 1-Methoxypropane | Ethoxyethane |
| | Tautomerism | $C_6H_6O$ | Oxepin | Benzene oxide |
| Stereo | Diastereomer (Geometric) | $C_2H_2Cl_2$ | cis-1,2-dichloroethene | trans-1,2-dichloroethene |
| | Enantiomer (Optical) | $C_3H_6O_3$ | (S)-Lactic acid | (R)-Lactic acid |

Optical activity is the phenomenon of shifting in the direction of the light plane when it passes through a compound. Light is an electromagnetic radiation that, when interacting with electrons in a molecule, slightly diverts its direction. Some molecules, though, do not present optical activity, i.e., for any light shift, either to the left or right, there is a molecule that shifts it in the opposite direction, nulling the optical activity. On the other hand, when light passes through pure there is a shift of the light; when light shifts to left, the enantiomer is called levorotatory (receives the prefix *l* or (−)) and to the right it is called dextrorotatory (receives the prefix *d* or (+)).

A polarimeter is the instrument responsible for reading the light rotation in degrees of a certain enantiomer concentration. This rotation must be converted into a specific rotation to take in account the polarimeter length and sample concentration, as given by:

$$[\alpha]_\lambda^T = \frac{\alpha}{l_{pol} \cdot c} \tag{1}$$

where $[\alpha]$ is the specific rotation, superscript $T$ designates the temperature (usually 20 °C), superscript $\lambda$ the light wavelength (usually at 589.6 nm of Na D-lines), $l_{pol}$ is the polarimeter length in dm, $c$ is the enantiomer concentration in g·cm$^{-3}$, and $\alpha$ is the observed rotation in degrees in the polarimeter. It means that an $l$ enantiomer has a negative specific rotation, a $d$ enantiomer a positive one, and a racemate has a $[\alpha]_\lambda^T$ of zero.

A pair of enantiomers does not overlap by either transposition or rotation. The non-overlapping of molecules is known in chemistry as chirality and these compounds are therefore called chirals (from the Greek word χέρια [quéria], meaning hands). In order to be considered chiral, the molecule must have at least one tetrahedral carbon bound to four different groups, as shown in Figure 1. This carbon is known as chiral or asymmetric carbon [4]. There is also the possibility of chirality involving other atoms such as silicon, nitrogen, sulfur and phosphorus, coordination complexes, allenes and atropisomers, as shown in Figure 2; however, these are less common [5].

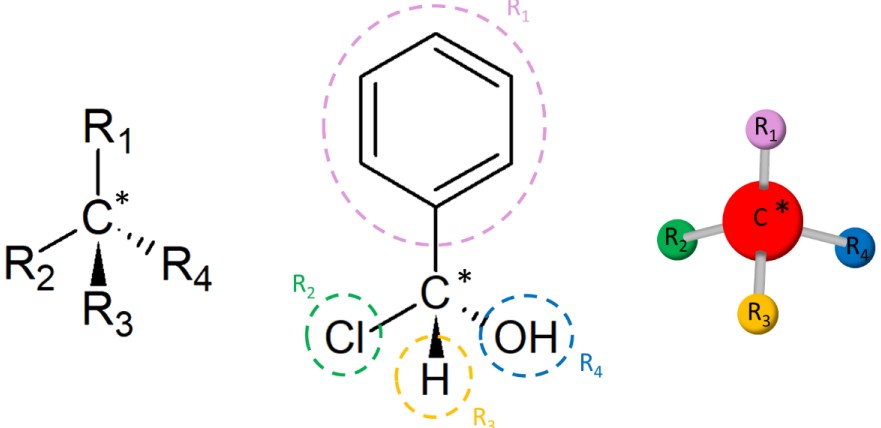

**Figure 1.** Chiral carbon (*) bound to four different groups in different representations.

Chirality in a molecule's structural formula is represented by solid and hashed wedges (◤ and ⋯⋯) representing bonds that are projected out towards and retreating from the viewer, respectively. At least two equal bonds make up the atom achiral; therefore it does not form optical isomers. The number of chiral carbons determines the number of enantiomers of a compound, as given by:

$$N_{EN} = 2^n \tag{2}$$

where $N_{EN}$ and $n$ are the number of enantiomers and chiral carbons, respectively, as exemplified in Figure 3. However, this correlation is not always true; molecules with at least two chiral carbons that present a plane of symmetry are not chiral molecules. One of its carbons shifts the light plan to left, whereas the other nulls its effect, an internal compensation. An example is illustrated in Figure 4, although the tartaric acid has two chiral carbons, it has only two enantiomers and one meso isomer, a diastereomer and a non-optically active stereoisomer (see Table 1).

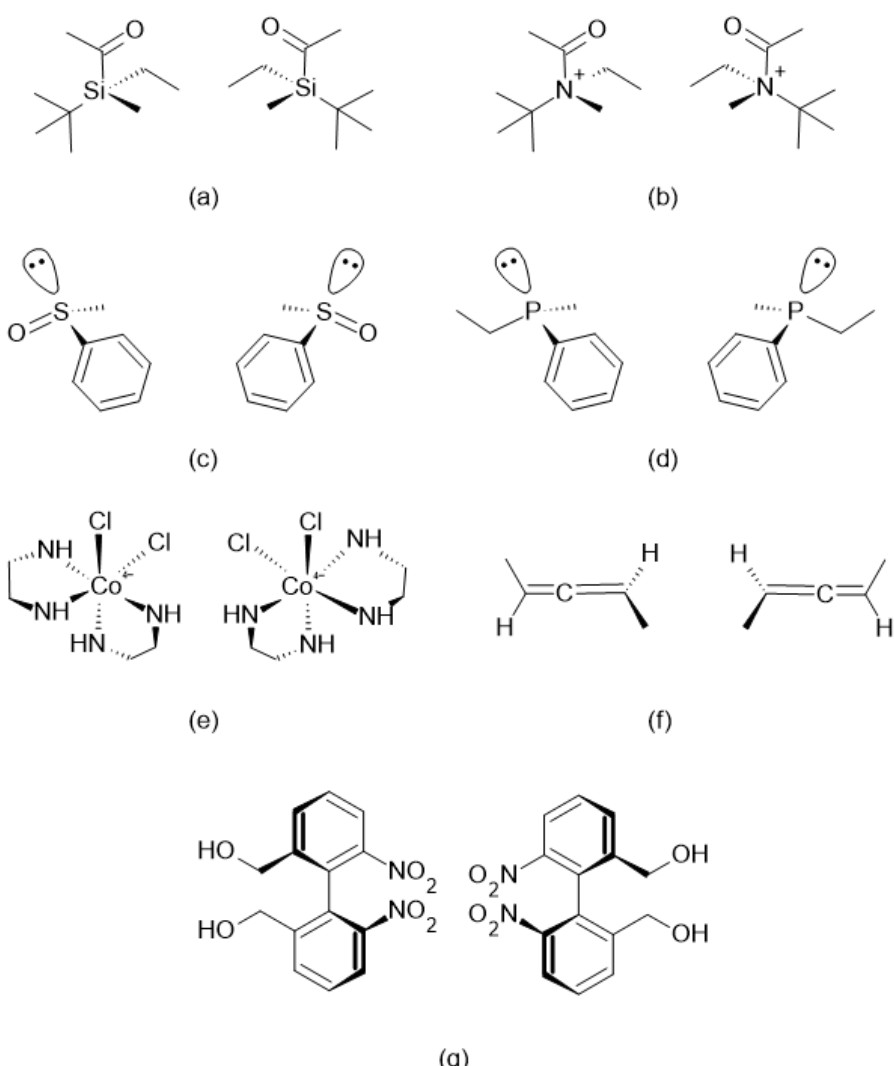

**Figure 2.** Pairs of enantiomers of (**a**) chiral silicon, (**b**) chiral nitrogen, (**c**) chiral sulfur, (**d**) chiral phosphorus, (**e**) cobalt complex, (**f**) allenes and (**g**) atropisomers.

Chiral carbons: $n = 1 \rightarrow$ Number of enantiomers: $2^n = 2^1 = 2$

(a)

Chiral carbons: $n = 2 \rightarrow$ Number of enantiomers: $2^n = 2^2 = 4$

(b)

**Figure 3.** Number of enantiomers related to the numbers of chiral carbons (*) for (**a**) lactic acid and (**b**) isoleucine.

**Figure 4.** Tartaric acid presents (**a**) two enantiomers and (**b**) a meso isomer.

## 2. Nomenclature

A pair of enantiomers is formed by two different compounds that must have two different names. The Chemical Abstracts Service Registry Numbers (CAS RN or CAS Number) is a registration system of different compounds. Chemical Abstract Service has registered over 190 million compounds since the XIX century and thousands are registered every day [6]. Its database includes inorganic and organic compounds; metals and alloys; polymers; isotopes; protein and nucleic acids; coordination compounds and organometallics; minerals; and even mixtures. In fact, not only the pure enantiomers, but racemates, a mixture of compounds, have their own CAS Number. For instance, limonene racemate's, (+)-limonene's and (−)-limonene's CAS Numbers are, respectively, 138-86-3, 5989-27-5 and 5989-54-8. The CAS Number does not have a significance itself, as it is just based on the order of registration. On the other hand, prefixes (*d* or +) and (*l or*−), mentioned in the previous section, designates whether the compound shifts the light to the right or left, and does not distinguish enantiomers when there are four or more enantiomers, as shown in Figure 3b; therefore, this is not a form of naming either.

Pairs of enantiomers are named following a sequence of rules created by the chemists R.S. Cahn, C. Ingold, and V. Prelog. This is known as Cahn-Ingold-Prelog rules or R/S configuration, where R and S stand for rectus and sinister (right and left in Latin). The rules for enantiomer nomenclature are detailed in Table 2.

First, it is necessary to rank the atoms directly bounded to the chiral carbon from largest to smallest atomic number, even in the case of isotopes. When the atoms have the same atomic number, one must follow down to the next substituent until there is a divergence point. Then, the least atomic number substituent is posed at the rear from a 3D perspective (a molecular modeling kit might help), as shown in Table 2. Finally, one draws circle arrows from the first to the third position. If the arrow goes counterclockwise, the enantiomer is S, otherwise is R. This procedure must be repeated when the molecule has other chiral carbons.

It is worth noticing that the prefixes (R) and (S) have no correlation with the shift of light plan, as an enantiomer might receive a prefix (R) and shifts light to left and vice versa. For instance: (R)-limonene has a specific rotation of +12° whereas (R)-butanol presents −13.5°. Another important issue that should be noted that there is another form of nomenclature exclusive for amino acids and monosaccharides, *L* and *D*, which have nothing to do with prefixes (R)/(S) or (*d*)/(*l*). *L*-amino acids and *D*-monosaccharides are by far the most common types that are present in all forms of life, with *D*-amino acids and *R*-monosaccharides nearly inexistent. They receive the prefix *L* when the amine or hydroxy group on the farthest carbon chiral from the carbonyl group is on the left side in a fisher projection, and prefix *D* when is on the right, as shown in Figure 5.

**Table 2.** Rules for enantiomer nomenclature.

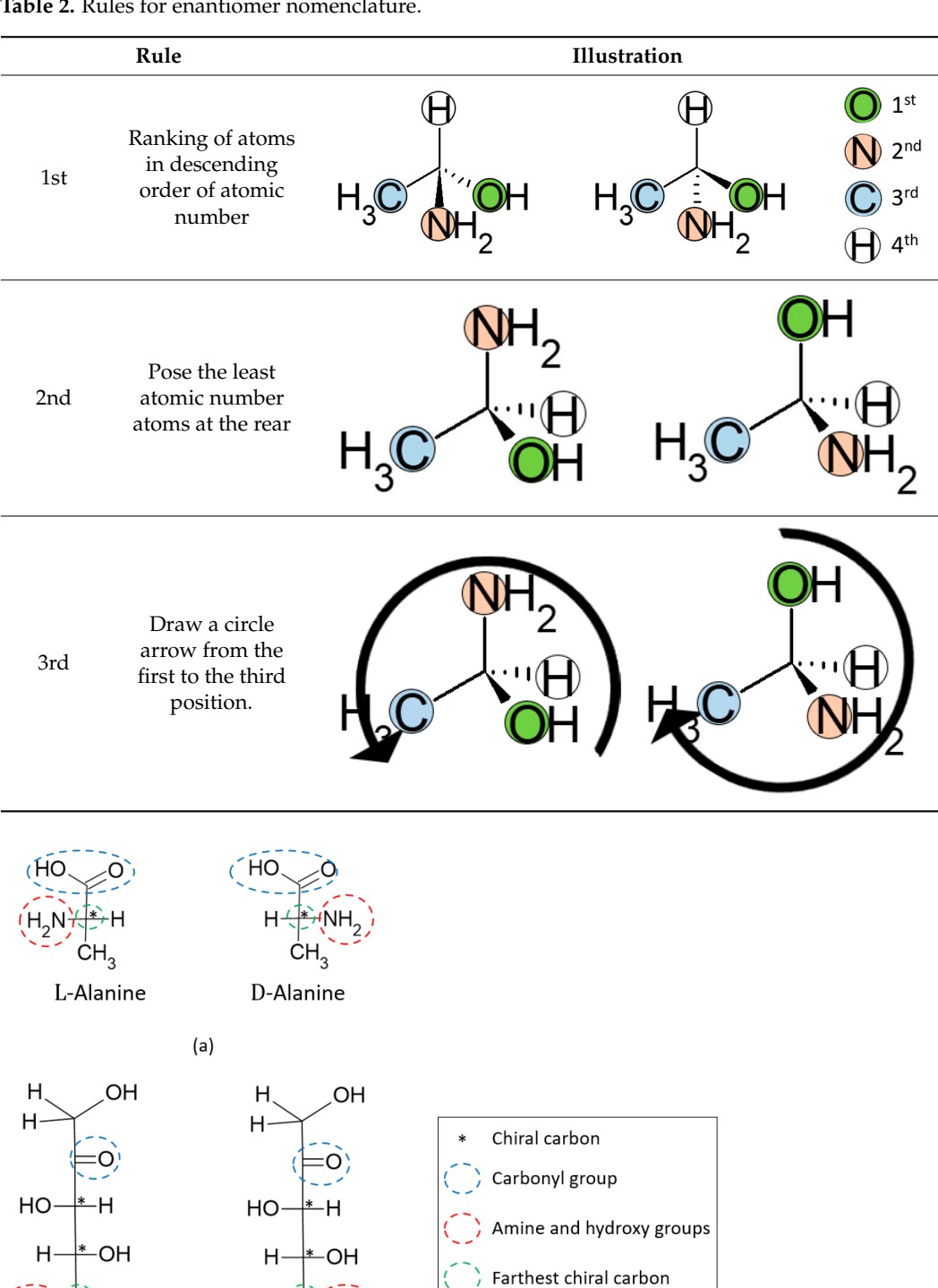

| Rule | | Illustration |
|------|---|--------------|

Figure 5. (**a**) Amino acids *L* and *D* -Alanine and (**b**) monosaccharides *L* and *D* -Fructose.

## 3. Enantiomers and the Human Body

Several natural compounds have two or more enantiomeric forms due to the complexity of their molecules that hold one or more chiral carbons. The human body, in turn, is essentially asymmetric; several organic molecules present chiral carbon, such as

deoxyribonucleic and ribonucleic acid and their monosaccharides deoxyribose and ribose, respectively, and monosaccharides of the natural polymer DNA and RNA, shown in Figure 6, are present in all human cells [7]. "Chirality represents an intrinsic property of the so-called 'structural blocks of life', such as amino acids and monosaccharides and, consequently, peptides, proteins and polysaccharides" [8].

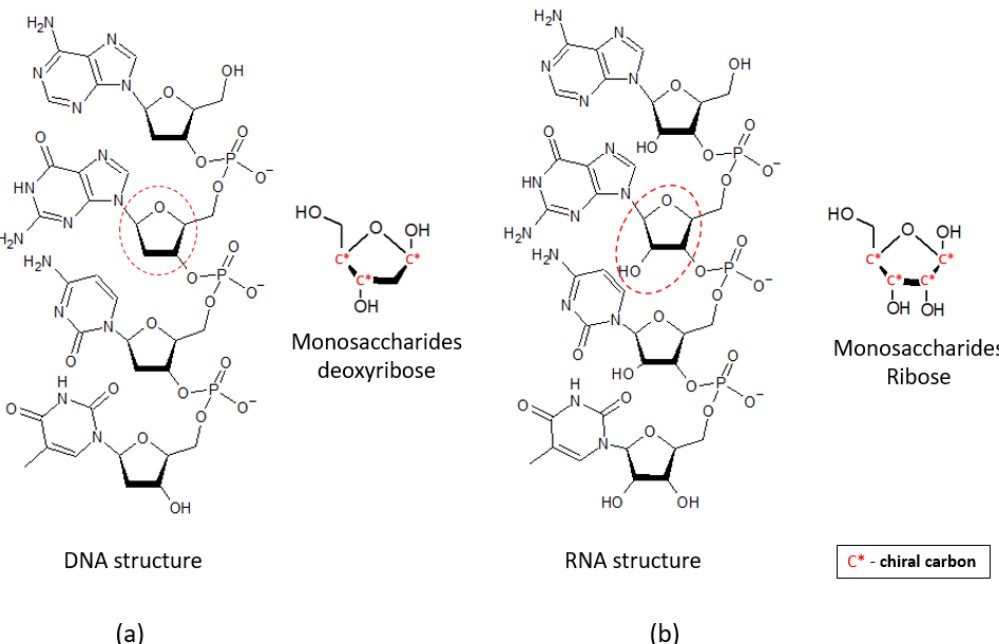

|          | DNA structure | RNA structure | C* - chiral carbon |
|----------|---------------|---------------|--------------------|
|          | (a)           | (b)           |                    |

**Figure 6.** Deoxyribose (**a**) and ribose (**b**) monosaccharides structural formulas from DNA and RNA and their chiral carbons.

In their composition, many drugs have enantiomeric compounds that, in an asymmetric environment such as the human body, act completely different. There are several cases where one of the enantiomer is the active principle, while the other might cause side effects, prove to be toxic, ineffective or antagonist in the treatment [2,9–12]. Many drugs must be sold in its pure enantiomer form in order to be effective and/or harmless, for this reason, enantioresolution is one of the most important issues of the pharmaceutical industry. There are many known cases in the literature of different behaviors of enantiomers in pharmaceutical, agronomic and food applications, as shown in Table 3 [4,13,14].

**Table 3.** Difference in biological activities between enantiomers [4].

| Compounds | (S) Enantiomer Actuation | (R) Enantiomer Actuation |
|-----------|--------------------------|--------------------------|
| Limonene  | Lemon odor               | Orange odor              |
| Carvone   | Caraway flavor           | Spearmint flavor         |

**Table 3.** *Cont.*

| Compounds | (S) Enantiomer Actuation | (R) Enantiomer Actuation |
|---|---|---|
| Asparagine |  Bitter taste |  Sweet taste |
| Aspartame |  Sweet taste |  Bitter taste |
| Ethambutol |  Tuberculostatic |  Causes blindness |
| Thalidomide |  Teratogen |  Sedative |
| Penicillamine |  Antiarthritic |  Mutagen |
| Ketamine |  Anesthetic |  Hallucinogen |
| Dopa |  Anti-Parkinson |  Serious side effects |

**Table 3.** *Cont.*

| Compounds | (S) Enantiomer Actuation | (R) Enantiomer Actuation |
|---|---|---|
| Chloramphenicol |  Inactive |  Antibacterial |
| Propranolol |  Antihypertensive, antiarrhythmic |  Contraceptive |
| Paclobutrazol |  Plant growth regulator |  Fungicide |

The reason for the distinct actuation of virtually identical compounds is found in the biological receptors in human cells. These are macromolecules present in cell membranes that mediate the effects of chemical messages, hormones, and drug actions in the body [13]. They are responsible for selecting which substance from extracellular fluid may or may not enter the cytoplasm [15]. Figure 7 illustrates this interaction between biological receptors and chiral molecules. In Figure 7a, the enantiomer interacts perfectly with the receptor, whereas in Figure 7b the connection is compromised, which might cause a side effect if any effect at all. The pharmacological properties, pharmacokinetic of adsorption, distribution, biotransformation and excretion, and drug toxicology of the enantiomeric drug must be well known and identified [8]. It is important to define the safe degree of purity for an enantiomeric drug before commercialization. Pharmaceutical companies are also interested in selling safe and efficient medicaments, especially in a highly competitive market of medicaments.

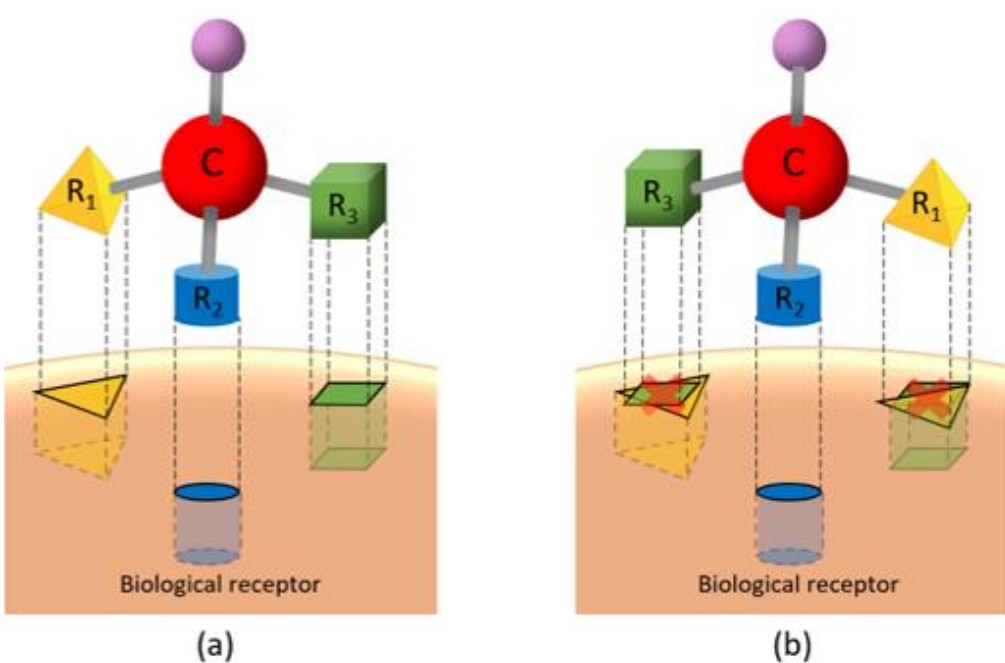

**Figure 7.** Perfect (**a**) and failed (**b**) connection between enantiomers and biological receptor.

## 4. Enantiomeric Drug Market

Nearly 60% of prescribed drugs have an enantiomeric pair [16,17]. However, for a long time, it was a common practice to commercialize chiral drugs in their racemic mixture. Regulatory authorities have put pressure on pharmaceutical companies to sell enantiomeric medicaments in their pure form [2,4]. Ethambutol was the first drug to be commercialized as a single enantiomer in 1961 [18]. Since then, the number of enantiomers sold in their pure form has increased year after year. In 1994, only 20% of the top selling drugs were enantiomers in their pure form [4]. In 2005 this number was already 37% [19] and currently it is over 50% [20]. This increase is justified to regulators that increasingly require clinical control and commercialization of pure enantiomers [8]. An example is the American Federal Agency FDA (Food and Drug Administration), which requires toxicological data of each enantiomer individually [4,21,22].

In economic terms, there was a worldwide growth of USD 30 billion to USD 100 billion from 1992 to 2000 with about 24 companies specializing in enantiomer separation [8,23]. In 2002, this value was already USD 159 billion and in 2005 USD 225 billion [8,19,24]. Table 4 shows the top pharmaceutical companies worldwide in 2017 with regard to chiral active ingredients, according to Pharmacompass (2019), which taken together account for almost USD 58.4 billion.

In such a market, pharmaceutical industries increasingly require refined compounds, which act quickly, in order to stand out in a competitive market. Due to the difference in the performance of different enantiomers, the complexity of some compounds, and the economic-market importance, it is crucial to efficiently and economically obtain each compound separately, either by synthesis or separation.

**Table 4.** Top seller drugs worldwide in 2017 with chiral active ingredients (PHARMACOM-PASS, 2019).

| Product Name | Chiral Active Ingredient | Indication | Revenue (in Millions of Dollars) |
|---|---|---|---|
| Revlimid | Lenalidomide | Oncology | 8187 |
| Xarelto | Rivaroxaban | Cardiovascular Diseases | 6590 |
| Lyrica | Pregabalin | Neurological/ Mental Disorders | 5317 |
| Imbruvica | Ibrutinib | Oncology | 4466 |

**Table 4.** *Cont.*

| Product Name | Chiral Active Ingredient | Indication | Revenue (in Millions of Dollars) |
|---|---|---|---|
| Harvoni |  Ledipasvir  Sofosbuvir | Infectious Diseases (HIV, Hepatitis, etc.) | 4370 |

**Table 4.** *Cont.*

| Product Name | Chiral Active Ingredient | Indication | Revenue (in Millions of Dollars) |
|---|---|---|---|
| Symbicort Pulmicort | Budesonide / Formoterol | Respiratory Disorders | 4360 |
| Januvia | Sitagliptin | Diabetes | 3737 |
| Epclusa | Sofosbuvir | Infectious Diseases (HIV, Hepatitis, etc.) | 3510 |

**Table 4.** *Cont.*

| Product Name | Chiral Active Ingredient | Indication | Revenue (in Millions of Dollars) |
|---|---|---|---|
| Triumeq | Abacavir<br><br>Dolutegravir<br><br>Lamivudine | Infectious Diseases (HIV, Hepatitis, etc.) | 3470 |
| Latuda | Lurasidone | Neurological/ Mental Disorders | 3350 |

**Table 4.** *Cont.*

| Product Name | Chiral Active Ingredient | Indication | Revenue (in Millions of Dollars) |
|---|---|---|---|
| Truvada |  Emtricitabine | Infectious Diseases (HIV, Hepatitis, etc.) | 3134 |
| Nexium |  Esomeprazol | Gastrointestinal Disorders | 2795 |
| Invega Sustenna |  Paliperidone Palmitate | Neurological/ Mental Disorders | 2569 |
| Zytiga |  Abiraterone Acetate | Oncology | 2505 |

## 5. Enantioresolution

Currently, there are two pathways to obtain chiral compounds in their pure form, namely, synthesis (chiral route) or separation (racemic route), as shown in Figure 8. In the first case, each enantiomer is produced separately by using a chiral catalyst that induces

selectivity for a given enantiomer, as shown in Figure 8a [4,8]. If the other enantiomer is also desired, it is necessary to develop a second synthesis with a different chiral catalyst. At first glance, the chiral route seems to be more advantageous than the racemic route (Figure 8b); they synthesize 100% of the intended enantiomer and there is no need for further separation and energetic expenses. On the other hand, racemic routes achieve up to 50% of the desired optically active isomer and racemic routes show inherently poor "atom economy", i.e., part of the raw material is wasted, and the methods are not "elegant" [25–28]. However, chiral routes usually have low overall yields and only a few of them are applied for industrial purpose, especially at the early stages of new drugs development that requires pure enantiomers for pharmacological tests [2]. Keith, Larrow and Jacobsen (2001) list the following conditions that must be met in order to make the chiral route feasible: cheap racemate; poor CSP (Chiral Stationary Phase) enantioselectivity; highly selective catalyst for one enantiomer; inexpensive or efficiently recyclable catalyst and economical and safe reaction. Usually, these conditions are difficult to achieve, in a way that the production of the racemic mixture for further separation (Figure 8b) is normally favored. Furthermore, economic interest is the driving force that boosts the development of new enantioresolution technologies [2]. The cost of separation depends on the desired degree of purity and this affects the separation route [29].

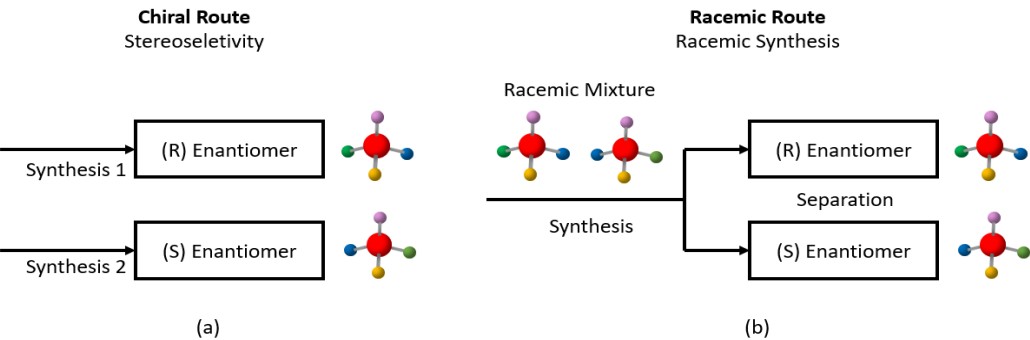

**Figure 8.** Enantiomers obtained by Chiral Route (**a**) Racemic Route (**b**).

Several methods for enantioresolution have been carried out in chiral environments using CSP. Chiral stationary phases are more effective at binding to one of the enantiomers, according to Dalgliesh's three-point rule [30], and analogous to biological receptor in Figure 7. This rule states that a chiral environment should have three interaction points with the retained compound either by H-bonding, electrostatic interaction, dipole stacking, inclusion complexion or steric bulk hindrance. When the enantiomer does not interact perfectly with the CSP, the connection weakens and the enantiomer is less retained in the chiral environment, promoting enantioseparation [31–33].

Enantioresolution methods have existed since the early works of Louis Pasteur, who was able to separate tartaric acid manually by crystallization [4]. Currently, there are several methods for enantioresolution, such as chromatography, crystallization, and membrane. Figure 9 summarizes the main methods for enantiomer separation that might be Crystallization, Membrane, and Chromatography. The latter can be divided into Gas Chromatography (GC), Supercritical Fluid Chromatography (SFC), and Liquid Chromatography (LC). The latter can be further classified into High Pressure Liquid Chromatography (HPLC), True Moving Bed (TMB) and Simulated Moving Bed (SMB) chromatography. All these methods have advantages and drawbacks and choosing the best one among numerous technologies is not straightforward; therefore, there is no single resolution method suitable for all racemate [8,29,34,35]. The main mechanisms for enantioresolution are going to be detailed in the following sections.

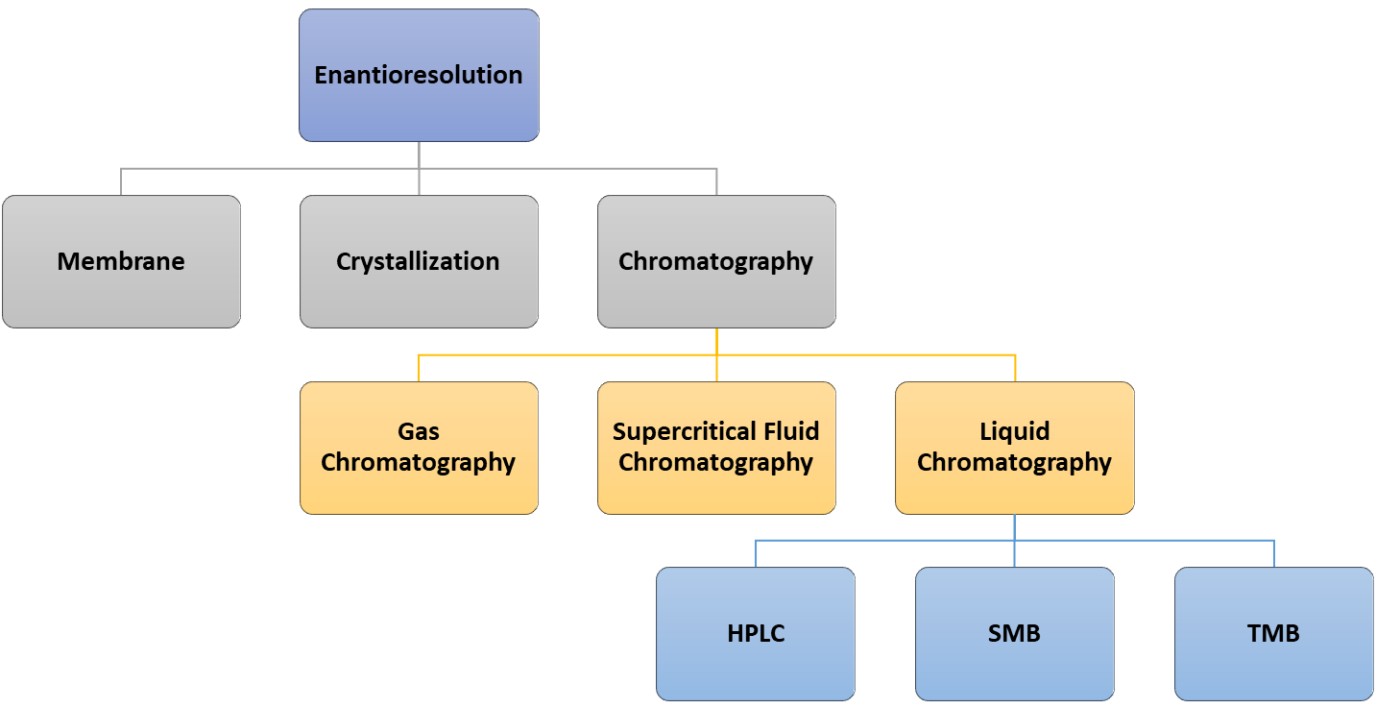

**Figure 9.** Main methods for enantiomer separation.

*5.1. Crystallization*

Crystallization is a stochastic phenomenon of molecular dynamics based on many variables such as temperature, pressure, glass-forming ability of liquids, among other not well understood [36]. Table 5 presents some examples of enantiomers resolved by crystallization methods in the literature. Crystals might be formed by either conglomerates or racemic crystals [37]. Conglomerates are mechanical mixtures of crystals macroscopically distinguished from the pure enantiomers. The first enantiomeric resolution was carried out by Louis Pasteur in 1848, who manually separated conglomerate crystals of sodium ammonium tartrate [37,38]. In racemic crystals, the two enantiomers crystallize together forming a one-phase crystal containing the same amounts of each enantiomer, as shown in Figure 10.

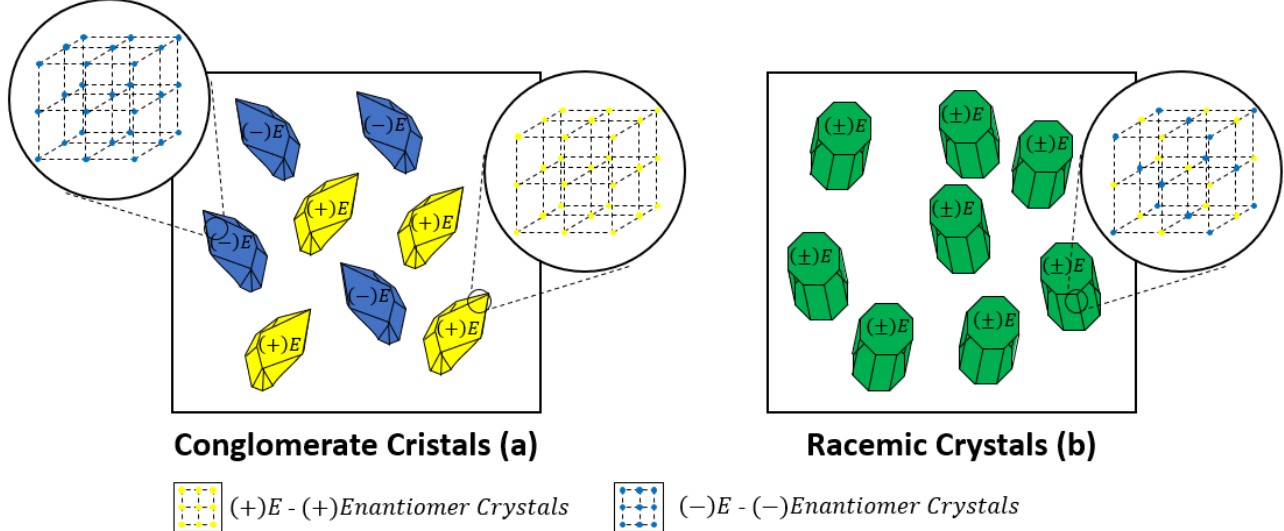

**Figure 10.** Illustration of conglomerate (**a**) and racemic compound (**b**) crystals.

Conglomerate crystal separations are generally straightforward; during the process of crystallization, enantiomers separate spontaneously into two stable crystal-phases, this mechanism is called preferential (or direct) crystallization and it is a non-expensive and efficient method, even on large scales [39,40]. Unfortunately, crystals from conglomerates are rather rare, comprising only 5–10% of all racemate crystals [38,40–42].

**Table 5.** Experimental studies of enantioresolutions applying crystallization.

| Enantiomers | Ref. |
| --- | --- |
| Ketoprofen | [36] |
| 5-ethyl-5-methylhydantoin | [39] |
| Threonine | [42] |
| Aspartic acid and glutamic acid | [43] |
| Propranolol | [44] |
| N-methylamphetamine | [45] |
| Threonine | [46] |
| Mandelic acid | [47] |
| Chiral microspheres based on poly(N-vinyl a-L-phenylalanine) | [48] |
| Benzo-(c)phenanthrene, 3,4-dehydroproline anhydride, and 2,6-dimethylglycoluril | [49] |
| 2-(2-oxopyrrolidin-1-yl)butanamide | [50] |
| Allenyl-bis-phosphine oxides | [51] |
| Leucine | [52] |
| Ibuprofen lysine | [53] |

Racemic crystals, in turn, require suitable resolving agents, which are optically active compounds that aid in their non-spontaneous separation. Resolving agents convert the enantiomeric pair into two different diastereomers (geometric isomers, see Table 1) with different Physico-chemical properties, as they are different compounds, such as solubility, thus, facilitating their separation [44,54]. This mechanism is the so-called Classical Resolution [38]. The main idea is to break the symmetry between enantiomers by adding a chiral agent in an achiral solvent, possibly forming non-covalent and covalent diastereomers. The former involves salt formation by adding acidic or basic substrates and is the most common resolution method and the latter is applied for molecules unable to form salt [40,54,55].

*5.2. Membrane*

Membranes for enantiomer separation have advantages over other methods because they tend to be low energy consuming, can work in continuous operation, even in large scales, can be scaled up and down, have higher throughput than other methods and are eco-friendly [31,40,56–58]. The membranes provide good a transport rate, high selectivity, stability in wide range of pH among different solvents and might be produced from different polymeric materials [59]. Enantioselective membranes work as barriers with chiral recognition sites that selectively transport one of the enantiomers based on affinity between the enantiomer and chiral selectors. Table 6 presents some examples of enantiomers resolved by membrane methods in the literature.

**Table 6.** Experimental studies of enantioresolutions applying membranes.

| Enantiomers | Membrane | Ref. |
|---|---|---|
| Phenylalanine | Immobilized DNA membranes | [57] |
| Phenylalanine | DNA-immobilized chitosan membranes | [58] |
| Phenylalanine | Polyaniline | [60] |
| Tryptophan, tyrosine, and phenylalanine | poly(γ-methyl-l-glutamate) membranes | [61] |
| Lactic acid and alanine | Polypropylene hollow-fiber module liquid membrane | [62] |
| Naproxen | Poly(4-vinylpyridine) /polypropylene membranes | [63] |
| Propranolol | Chiral derivatized polysulfone | [64] |
| Tryptophan, henylglycine and phenylalanine | Immobilized DNA membranes | [65] |
| 1-phenylethanol | (R,R)-TADDOL | [66] |
| N-protected amino acid derivatives | Adamantyl-carbamoyl-11-octadecylthioether-quinine/-quinidine | [67] |
| Tryptophan | Chitosan/-cyclodextrin composite membranes | [68] |
| Tryptophan | Cellulose dialysis membranes | [69] |
| 2-phenyl-1-propanol | Glutaraldehyde-crosslinked chitosan membranes | [70] |
| Tryptophan | BSA-Immobilized and BSA-Interpenetrating Network Polysulfone Membranes | [71] |
| Ketoconazole | Hydrophobic l-isopentyl tartrate and hydrophilic sulfobutylether–cyclodextrin | [72] |
| Amlodipine | Hollow fiber supported liquid membrane | [73] |
| Phenylalanine | Hollow fiber supported liquid membrane | [74] |
| Atenolol | Nano-sized chiral imprinted polymers | [75] |
| Ibuprofen | L-tartaric acid derivatives | [76] |
| DOPA | L-Glutamic acid-Graphene oxide based membranes | [77] |
| Tyrosine, phenylalanine and tryptophan | D-penicillamine-modified membrane and N-acetyl-L-cysteine-modified membrane | [78] |
| Phenylalanine | Regenerated cellulose membranes | [79] |
| Arginine | Chiral channel protein (FhuAF4) | [80] |
| Baclofen | Silica-based vancomycin-chiral stationary phase | [81] |
| Methadone | Chiral (2-hydroxypropyl)-β-cyclodextrin | [82] |

Enantioselective membranes might be either liquid or solid [2,56]. Liquid membranes are formed by organic liquid with Chiral Selectors (CS) dissolved or suspended in it, such as cyclodextrins, crown ethers, chiral copper complexes, DNA, polypeptides, and enzymes. Solid membranes for enantiomer separation use chiral polymers as enantioselective compounds that interact to a specific enantiomer, as shown in Figure 11 [62]. They are either matrices of molecularly imprinted chiral polymers or membranes with the chiral polymeric selector immobilized (by impregnation, esterification, or grafting) on porous membranes [33,40,56,68,71].

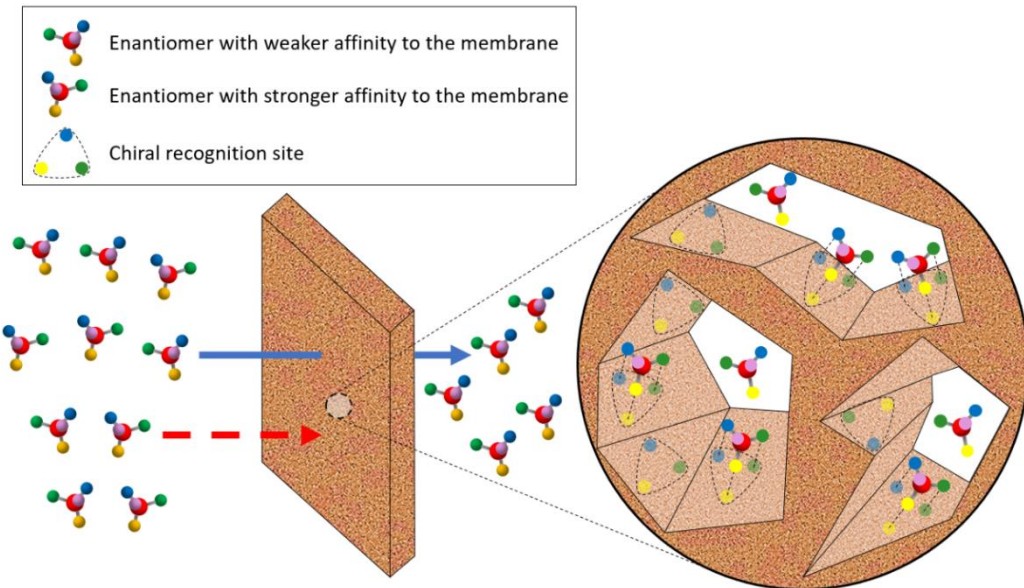

**Figure 11.** Chiral recognition sites in a porous solid membrane selecting a specific enantiomer.

*5.3. Chromatography*

Chromatographic processes are largely applied in several fields such as chemistry, pharmacy, and bioprocesses either for purification or recovery of different compounds [83]. Its separation is based on a physico-chemical phenomenon whereby a compound in a mobile phase known as the eluent, liquid or gaseous phase, is adsorbed onto the surface of a solid phase, usually a porous adsorbent. Chromatography displays some advantages over crystallization and membrane separations once it may be applied for a mixture of more than two chiral compounds and it can run with samples of small amount for analytical purposes [84]. Its separation is based on the chemical affinity of each compound to a stationary phase. In a chromatography equipment there are a solid stationary phase, generally porous, and a liquid or gaseous (solvent) mobile phase, which contains the mixture to be separated. Chromatographic methods have been the most effective for obtaining enantiomers with very high purity [4,8].

There are different chromatographic techniques for chiral resolution that are able to separate virtually all racemic compounds, that includes GC, SFC, and LC, as shown in Figure 9. Regardless of the chromatographic method applied, a Chiral Stationary Phase is applied, according to Dalgliesh's three-point rule [30]. Figure 12 depicts adsorption of Enantiomer 1 (E1) onto chiral environment, following Dalgliesh's three-point rules, whereas Enantiomer 2 (E2) has a compromised interaction to the surface, not fixing on it. CSPs are adsorbents usually based on cellulose such as the commercial products Chiralcel OD, CHIRALCEL OZ, CHIRALCEL OJ and CHIRALPAK AD, whereas common eluents are hexane, ethanol, and methanol. The enantiomers commonly resolved are pharmaceutical drugs such as Praziquantel and Guaifenesin.

All chromatographic techniques addressed to enatioresolution have their advantages and disadvantages. The next sections detail the differences among these chromatographic techniques, their functionalities and particularities.

### 5.3.1. GC—Gas Chromatography

Gas chromatography can be used by indirect or direct mechanisms [85]. In the former, enantiomers are converted into diastereomers by a resolving agent, analogous to a crystallization in racemic crystals, then they are separated in an achiral filling CG. The latter uses CSP that separates one of the enantiomers preferentially in a straightforward way by means of chemical affinity based on the three-point rule dismissing a pretreatment [86]. For

the direct mechanism, GC requires the use of volatile and thermally stable compounds and the choice for a CSP is of paramount importance in order to resolve racemic mixtures [87].

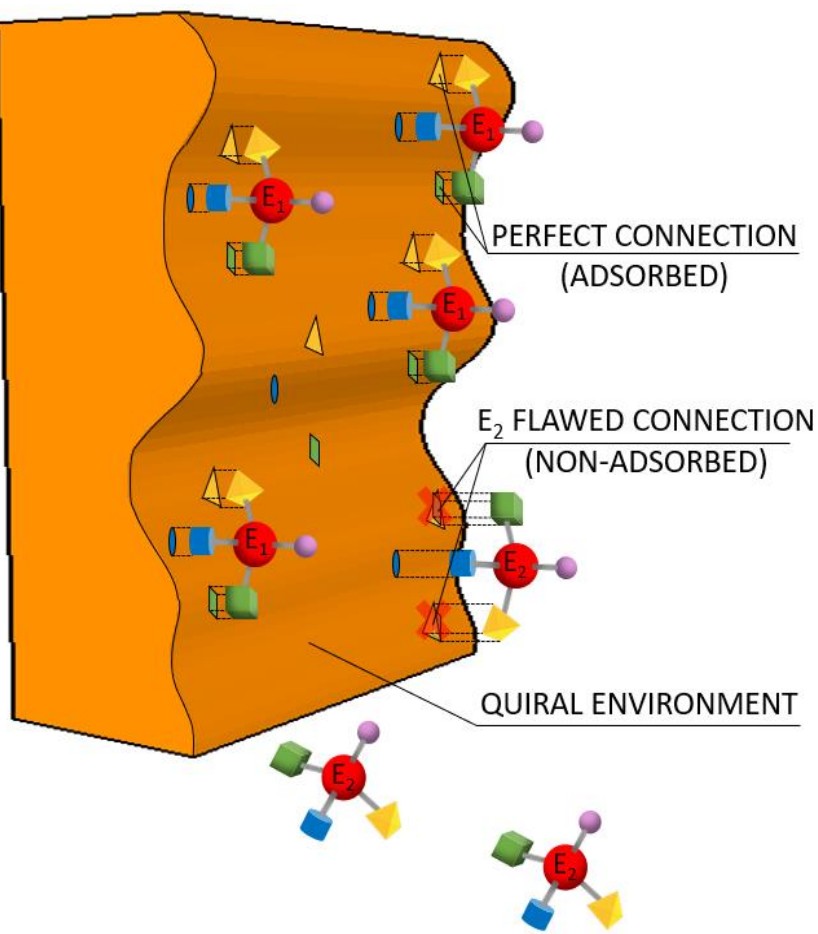

**Figure 12.** Adsorbed Enantiomer 1 (E1) onto the chiral environment according to Dalgliesh's three-point rule and non-absorbed Enantiomer 2 (E2).

The first GC with a Chiral Stationary Phase was applied to resolve 2-nalkanols [85]. Since this achievement, GC has been extensively used in academia and industry due to its advantages of being high efficient, simple, and sensitive in comparison to other methods [85,86,88,89]. On the other hand, GC has the drawbacks of being very difficult to scale up and their CSPs usually racemize, decompose and bleed at high temperatures, thereby diminishing its separation factor [87,90]. Table 7 presents some examples of enantiomers resolved by gas chromatography methods.

**Table 7.** Experimental studies of enantioresolutions applying gas chromatography.

| Enantiomers | CSP | Gas Carrier | Ref. |
|---|---|---|---|
| β-Blockers | (-)-α-methoxy-α-(trifluoromethyl) phenylacetyl chloride | Helium | [91] |
| β-pinene, sabinene, limonene, linalool, terpinen-4-ol, α-terpineol, linalyl acetate | EtTBS-βCD and DB-5 | Hydrogen | [92] |
| Chiral alcohols, chiral sulfoxides, chiral epoxides and acetylated amines | Chiral ionic liquid stationary phases | Helium | [93] |
| Flurbiprofen and ketoprofen | Agilent 6890 gas chromatograph | Helium | [94] |

**Table 7.** *Cont.*

| Enantiomers | CSP | Gas Carrier | Ref. |
|---|---|---|---|
| Alanina, prolina, serina, asparagine, glutamine, lisine, ornitina | Chirasil-l-Val capillary columns | Helium | [95] |
| Obuprofen, fenoprofen and ketoprofen methyl esters | Heptakis-(2,3-di-Omethyl-6-O-t-butyldimethyl-silyl)-β-cyclodextrin | Hydrogen | [96] |
| Chiral epoxides | Cyclodextrin derivatives | Nitrogen | [97] |
| α-amino acids | Modified Linear Dextrins | Hydrogen | [98] |
| Methyl branched compounds | 2,3-Di-O-methoxymethyl-6-O-tert-butyldimethylsilyl-γ-cyclodextrin | Hydrogen | [99] |
| Hydrocarbons, underivatized alcohols, ketones, and proteinogenic amino acid derivatives | Permethylated-βcyclodextrin and resorcinarene with pendant L- or D-valine diamide groups | Hydrogen | [100] |
| β-Blockers | DB-5 and DB-17 dual-columns | Helium | [101] |
| 2,2-dimethylcyclopropane-carboxamide | g-cyclodextrin | Helium | [102] |
| 12 amino acids | N-Ethoxycarbonylation was combined with (S)-1-phenylethylamidation | Helium | [103] |
| β-amino acid | CP-Chirasil-Dex CB and CP-Chirasil L-Val | Nitrogen | [104] |
| 1-phenylethanol | Permethylated -cyclodextrin | Nitrogen | [105] |
| 3-methylhexane, 2,3-dimethylpentane, 3-methyl-heptane, 3,4-dimethylhexane, 2,4-dimethylhexane, 2,3-dimethylhexane, 2,2,3-trimethylpentane | octakis(6-O-methyl-2,3-di-O-pentyl)-g-cyclodextrin | Hydrogen | [106] |
| Amino acid derivatives | (l)- or (d)-Valine tert-butylamide grafted on permethylated -cyclodextrin | Helium | [107] |
| 2,4-dimethylhexane | octakis(6-O-methyl-2,3-di-O-pentyl)-γ-cyclodextrin | Nitrogen | [108] |
| α- and β-pinene, cis- and trans-pinane, 2,3 butanediol, γ-valerolacton, 1-phenylethyl-lamine, 1-phenylethanol, 2-ethyl-exanoic acid | Derivatized cyclodextrins | Helium | [109] |
| Methylamphetamine | γ-cyclodextrin | Helium | [110] |
| Citronellal, camphor, alanine, leucine, valine, isoleucine, 1-phenyl-1,2-ethandiol, phenylsuccinic acid, and 1-phenyl-ethanol | Chiral Metal-Organic Frameworks | Nitrogen | [111] |
| Cathinone- and amphetamine-related designer drugs | Trifluoroacetyl-l-prolyl chloride | Helium | [112] |
| Galaxolide, tonalide, phantolide, traseolide and cashmeran | Chiral heptakis(2,3- di-O-methyl-6-O-t-butyl dimethylsilyl)–cyclodextrin | Helium | [113] |
| Citronellal, 1-phenyl-1,2-ethandiol, 1-Phenyl-ethanol, 2-amino-1-butanol, limonene, methionine, proline | Porous Chiral Metal-Organic Framework | Nitrogen | [114] |
| 2-hexanol, linalool, citronellal, methyl l-b-hydroxyisobutyrate, limonene, rose oxide, dihydrocarvyl acetate, menthol, valine, and leucine | Metal–Organic Framework on a Chiral Cyclodextrin | Nitrogen | [115] |
| 30 amino acids | Press-Tight© connected Varian-Chrompack Chirasil-l-Val | Helium | [116] |

### 5.3.2. SFC—Supercritical Fluid Chromatography

Supercritical fluids display transient properties between liquid and gas phases; therefore SFC works as an intermediate between GC and LC [117]. At the supercritical condition, the substances are above its critical temperature and pressure [118,119]. Supercritical fluid was firstly used as an eluent for chromatographic separation in 1962 by Klesper, Iiber and Clark using chlorofluoromethanes at 140 bar and 150–170 °C [120,121]. Supercritical fluids have advantages over liquid and gas states and they are a better solvent due to their higher density in comparison to gas states as they are faster (shorter run times), require lower pressure across column and require lower volume due to its lower viscosity and higher diffusivity over liquid states [118–120,122–127]. The most commonly applied supercritical fluid is carbon dioxide; however, the mentioned features are not exclusive to fluids over the critical point, and equal properties are seen in subcritical regions, since there are no abrupt changes of properties in the transition to a supercritical phase. This means that the fluid must not be at a supercritical condition to perform an SFC [118]. Table 8 presents some examples of enantiomers resolved by SFC methods reported in the literature.

**Table 8.** Experimental studies of enantioresolutions applying Supercritical Fluid Chromatography.

| Enantiomers | CSP | Ref. |
|---|---|---|
| Ibuprofen | Kromasil CHI-TBB, Kromasil CHI-DMB, Chirobiotik T, Chiracel OBH and Chiralpal AD | [128] |
| Dioxolane compounds | Chiralpak AD and Chiralcel OD | [129] |
| Enantiomeric pharmaceuticals | Chiralpak AD | [130] |
| Enantiomeric pharmaceuticals | Chiralcel OD, Chiralcel OJ and Chiralcel AD | [131] |
| A set of 111 chiral compounds | Chirobiotic T, Chirobiotic TAG and Chirobiotic R | [132] |
| Albendazole sulfoxide | Chiralpak AD and Chiralcel OD | [133] |
| Triadimefon and triadimenol | Chiralpak AD | [134] |
| Albendazole sulfoxide | Chiralpak AD | [135] |
| Omeprazole and several related benzimidazoles | Chiralpak AD | [136] |
| Triazole pesticides | Chiralpak AD | [137] |
| Enantiomeric pharmaceuticals | Chirlapak AD and AS, and Chiralcel OD and OJ | [138] |
| Antiulcer drugs | Chiralpak AD | [139] |
| Naproxen | Kromasil CHI-TBB | [140] |
| Warfarin | Chiralpak AD-H | [141] |
| Chiral sulfoxides | Chiralpak AD | [142] |
| Antimycotic azole drugs | Chiralpak AD | [143] |
| Nutlin-3 | Chiralcel OD, Chiralcel AD, Chiralcel OJ, Chirobiotic T, Chirobiotic V | [144] |
| Phospine-Containing α-Amino Acid Derivatives | Lux Cellulose-1 and -2 | [145] |
| Acetamide intermediate | Chiralcel OD-H, Chiralpak AD, Lux Cellulose-2 and Lux Amylose-2 | [146] |
| Tris-(3,5-dimethylphenylcarbamate) of amylose | Chiralcel OD-H and Chiralpak AD-H | [147] |
| Mianserin | Chiralcel OJ | [148] |

**Table 8.** *Cont.*

| Enantiomers | CSP | Ref. |
|---|---|---|
| Chiral fluoro-oxoindole-type compounds | Lux Cellulose-1, Lux Cellulose-2 and Lux Amylose-2 | [149] |
| Flutriafol | Chiralpak IA-3 | [150] |
| Enantiomeric pharmaceuticals | Chiralpak IC and Chiralpak AD-3 | [151] |
| Troeger's base, binaphthol, mandelic methylester, trans-stilbene oxide, flavanone and guaifenesine | Chiralcel AD-H and Chiralpak IC | [152] |

Additionally, $CO_2$ is the most used eluent in SFC due to its relatively low critical point (304.12 K and 73 atm), and it can be purified and reused after analysis; moreover it is inert, non-toxic, non-flammable cheap and relatively safe gas, therefore being considered a green solvent [118,119]. Due to its low critical temperature point, operating $CO_2$ in SFC reduces the likelihood of CSP racemization, improving enantioselectivity [124,153].

5.3.3. LC—Liquid Chromatography

Liquid chromatography has been growing in importance in the last decade due to the variety of enantioseparation modalities [5,154,155]. In LC, the eluent and compounds cross the fixed bed column (adsorber) by means of gravity and this can prove to be lengthy in some procedures. The time-consumption of LC can be overcome by means of a pump, low-dead-volume injectors, and detectors. An enhancement of LC is the so-called High-Performance Liquid Chromatography (HPLC—also known as High-Pressure Liquid Chromatography). HPLC is suitable for a wide range of applications, such as pharmaceuticals and food analysis. HPLC has the advantage of running a fast analysis and high resolution; however, in HPLC (and LC) the adsorber must be constantly regenerated, once it will be contaminated by the more retained enantiomer. Table 9 presents some examples of enantiomers resolved by HPLC methods.

**Table 9.** Experimental studies of enantioresolutions applying HPLC Chromatography.

| Enantiomers | CSP | Eluent | Ref. |
|---|---|---|---|
| Antifungal chiral drugs | Polysaccharide derivatives | Hexane-ethanol and hexane-2-propanol | [133] |
| Fungicide Enantiomers | Amylopectin Based Chiral | n-hexane and isopropanol | [144] |
| Oxazepam, lorazepam, and temazepam | Derivatized cyclodextrin-bonded | Acetonitrile | [156] |
| 1,4-Dihydropyridines | Vancomycin | Methanol/acetic acid/TEA | [157] |
| Linezolid | Amylose based | Hexane, 2-propanol and trifluoro acetic acid | [158] |
| Tangutorine | Chiralcel OD and Chiralpak AD | n-hexane/2-propanol | [159] |
| β-blockers | (R)-1-naphthylglycine and 3,5-dinitrobenzoic acid | n-hexane, 1,2-dichloroethane and methanol | [160] |
| Tolterodine tartarate | Chiralcel OD-H | n-hexane and isopropyl | [161] |
| 1,4-dihydropyridinemonocarboxylic acid | Tert-butylcarbamoylquinine | Methanol and ammonium acetate buffer | [162] |
| Naringenin and other flavanones | Chiralcel OD-H and Chiralpak AS-H | n-hexane/alcohol | [163] |
| Piperidine-2,6-dione analogues | Chiralpak IA and Chiralpak IB | Methyl-tert-butyl ether-THF | [164] |
| Bambuterol | Chiralpak AD | Hexane/2-propanol | [165] |

**Table 9.** *Cont.*

| Enantiomers | CSP | Eluent | Ref. |
|---|---|---|---|
| β-Lactams | Cyclodextrin-Based Chiral | Isopropanol-heptane | [166] |
| Ruthenium(II) Polypyridyl Complexes | Cyclodextrin Chiral | Methanol and acetonitrile | [167] |
| Chiral acids, bases, and amino acids | Zwitterionic ion-exchange-type | Acetic acid, formic acid, diethylamine, and ammonium acetate | [168] |
| 10 β-adrenergic blockers | CelluCoat column | n-heptane–ethanol–diethylamine | [169] |
| Triazole Fungicides | Chrialcel OD and Chrialcel OJ | Hexane/2-propanol | [170] |
| 2-arylpropionic acid nonsteroidal anti-infl ammatory drugs | Hydroxypropyl-β-cyclodextri | Methanol and $NaH_2PO_4$ buffer | [171] |
| 4 β-adrenergic blockers | SPE-Chiral | n-Heptane:ethanol:diethylamine | [172] |
| Ofloxacin | Ionic liquid-assisted ligand-exchange | Methanol/water | [173] |
| Chiral Pesticides | Cellulose tris-(3,5-dimethylphenyl-carbamate)-coated chiral | Ethanol, n-propanol, iso-propanol, n-butanol, and iso-butanol | [174] |
| Dihydropyridine derivatives | Polysaccharide-based chiral | Formic acid | [175] |
| Arylpropionic acid derivatives | Chiralpak AD | n-hexane modified either with 2-propanol or ethanol | [176] |
| Illicit drugs | Cyclofructan-based and cyclobond I 2000 RSP | Heptane with ethanol or isopropanol | [177] |
| Ruthenium (II) Polypyridyl Complexes | Cyclofructan | Acetonitrile and methanol | [178] |

True Moving Bed (TMB) is an attempt to develop a continuous liquid chromatography based on the counter-current movement of eluent and solid. Its separation is maximized by a constant flow of solid and liquid phases counter-currently. In this chromatography, the CSP adsorber retaines one of the compounds due to its enantioselectivity nature. This leads to different velocities of displacement of enantiomers along the column. The more retained enantiomer takes longer to reach the end of the column than the less retained compound and, for better separation, the column must be long enough [83]. In TMB chromatography, the adsorber is constantly being regenerated by the eluent. In TMB chromatography, there is constant liquid and solid recirculation, as shown in Figure 13. The liquid stream leaves the top of the column at Section IV and is recycled to Section I, while the solid stream (adsorbent) moves in the opposite direction, being recycled from Section I to Section IV. The eluent and racemate feed the system (respectively, streams E and F in Figure 13). As shown in Figure 13, the more retained compound leaves the system in the Extract Stream (X), whereas the less retained compound is removed in the Raffinate Stream (R). The system has four sections with different functions [83], as explained in Table 10. Due to the counter-current flows, TMB should reach higher purity, even if the adsorbent (solid phase) presents low selectivity, in contrast to conventional chromatography where high selectivity is crucial [83]. However, from an engineering perspective, solid movement is difficult to attain and can cause mechanical erosion in the adsorbent phase, equipment abrasion, and difficulties in maintaining plug flow for the solid [4,83]. The TMB system is a theoretical concept and to solve such problems, an SMB chromatographic unit was developed.

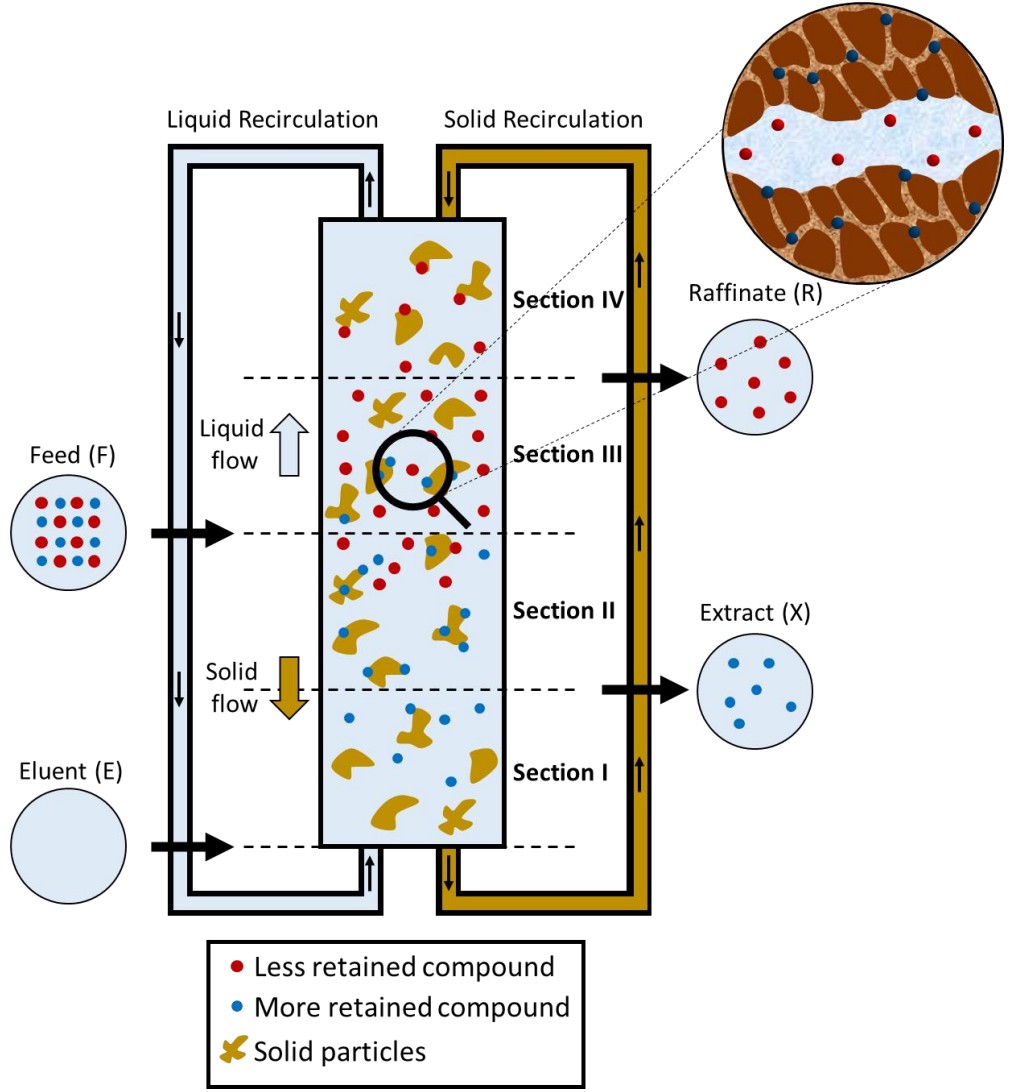

**Figure 13.** Schematic diagram of a TMB chromatograph.

**Table 10.** Functions of the TMB sections.

| Section | Function | Illustration |
|---|---|---|
| I | The more retained compound moves upward desorbed with the eluent, so that it leaves the system in the Extract stream (X). The eluent cleans the solid that is regenerated prior to being recycled in Section IV. | 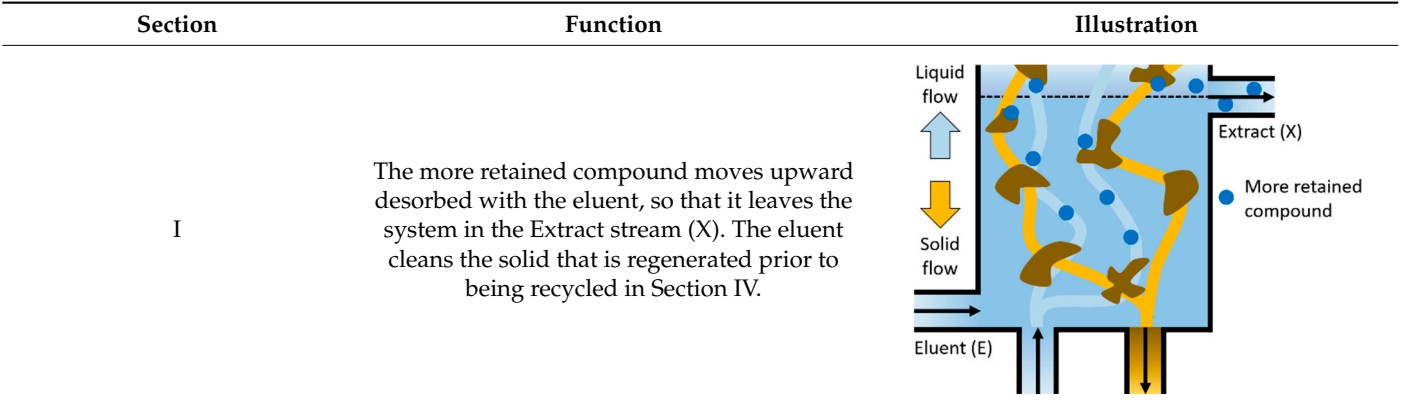 |

**Table 10.** *Cont.*

| Section | Function | Illustration |
|---|---|---|
| II | The more retained compound moves downward and is adsorbed on the solid whereas the less retained compound is desorbed with the eluent. This prevents contamination of the less retained compound in the Extract stream (X); the less retained compound moves upward to the Raffinate stream (R). |  |
| III | The more retained compound moves downward adsorbed on the solid and the less retained compound is desorbed with the eluent. This prevents contamination in the Raffinate stream (R); the more retained compound moves downward to the Extract steam (X). |  |
| IV | The less retained compound moves downward adsorbed with the solid flow, so that it leaves the system in the Raffinate stream (R). The solid phase cleans the liquid that is regenerated prior to be recycled to Section I. |  |

A Simulated Moving Bed is a powerful technology for the preparative and analytical scale in laboratory or in industry [83]. SMB chromatography was first presented by UOP (Universal Oil Products) through a United States Patent [4,179–181], known as the Sorbex Process, which was designed for oil refining purposes [4]. The technology soon found other industrial applications, such as in biotechnology, pharmaceutical and fine chemistry, as it is a separation system with advantages over batch chromatographic systems and traditional processes (PAIS, 1999). Negawa and Shoji (1992) were the first authors to carry out an enantioresolution in an SMB chromatography, they resolved 1-phenylethanol on CHIRALCEL OD as CSP. Since then, there have been several enantioresolutions reported in the literature, as summarized in Table 11.

**Table 11.** Experimental studies of enantioresolutions applying SMB chromatography.

| Enantiomers | CSP | Eluent | Ref. |
|---|---|---|---|
| Tramadol | CHIRALPAK AD 20 | 2-propanol, hexane and diethylamine | [182] |
| EMD 53986 | Cellulose-tri-(p-methyl-benzoate) and polymeric silica based | Ethylacetate and ethanol | [183] |
| Guaifenesin, aminoglutetimida, and formoterol | CHIRALCEL OJ and CHIRALCEL OD | Heptane/ethanol | [184] |
| 1,1′-bi-2-naphthol | 3,5-dinitrobenzoyl phenylglycine bonded to silica gel | Heptane and isopropanol | [185] |
| 1,1′-bi-2-naphthol | 3,5-dinitrobenzoyl phenylglycine bonded to silica gel | Heptane and isopropanol | [186] |
| Guaifenesin | CHIRALCEL OD | Heptane/ethanol | [187] |
| 1-phenoxy-2-propanol | CHIRALCEL OD | Hexane and isopropanol | [188] |
| 1,1′-bi-2-naphthol | 3,5-dinitrobenzoyl phenylglycine bonded to silica gel | Heptane and isopropanol | [189] |
| 1-phenyl-1-propanol | CHIRACEL OB | Ethyl acetate and heptane | [190] |
| Trans-stilbene oxide and Tröger's Base | CHIRALPAK AS and CHIRALPAK AS-V | Hexane/isopropanol | [191] |
| N-carbobenzoxy-tert-leucine and N-Boc-tert-leucine-benzylester | CHIRALCEL OD and CHIRALPAK AD | Heptane/ethanol and Heptane/2-propanol | [192] |
| Phenylpropanolamine | CHIRALPAK AD | Methanol | [193] |
| Guaifenesin | CHIRALCEL OD | Ethanol | [194] |
| Bupivacaine | Kromasil CHI-TBB | Iso-propanol, hexane, and acetic acid | [195] |
| DL-methionine | Eremomycin | Methanol and water | [196] |
| Tröger's base | CHIRALPAK AD | Methanol | [197] |
| α-Tetralol | CHIRALPAK AD | Heptane/2-propanol | [198] |
| (RS,RS)-2-(2,4-difluorophenyl)butane-1,2,3-triol | CHIRALCEL OJ and CHIRALPAK AD | Hexane, ethanol, and methanol | [199] |
| Tröger's base | CHIRALPAK AD | Ethanol | [200] |
| Mandelic acid | Kromasil TBB | Hexane and ter-butylmethylether | [201] |
| Tröger's Base | CHIRALPAK AD | Ethanol | [202] |
| Tröger's Base | CHIRALPAK AD | Ethanol | [203] |
| Ketoprofen | Chiralpak AD1 | Ethanol, hexane and trifluoroacetic acid | [204] |
| Ketoprofen | Chiralpak AD1 | Ethanol, hexane and trifluoroacetic acid | [205] |
| Guaifenesin | CHIRALCEL OD | Hexane/ethanol | [206] |
| Praziquantel | Chiralcel OZ | Methanol | [207] |

SMB simulates TMB counter-current flows and overcomes its intrinsic solid movement problems by keeping the solid phase fixed and switching the input and output streams cyclically and periodically (time switch) in a multi-column system, as shown in Figure 14, by simulating a bed movement [208]. SMB has four sections for which the function is analogous to TMB, as shown in Figure 13. The SMB system is fed with an eluent (stream E in Figure 14) and racemate to be separated (stream F in Figure 14). Two outlet streams remove the more and the less retained compounds, respectively, in the extract and raffinate streams (streams X and R, respectively in Figure 14). At the time instant $t_1$, the inlet and outlet streams are set as shown in Figure 14. At time $t_2$, they move clockwise from their current positions to the next ones. Next, they proceed to $t_3$ and so on until they return to their original positions, thus finishing the cycle. The separation is promoted by adsorption of the enantiomer with greater chemical affinity onto the porous solid phase, as zoomed in Figure 14.

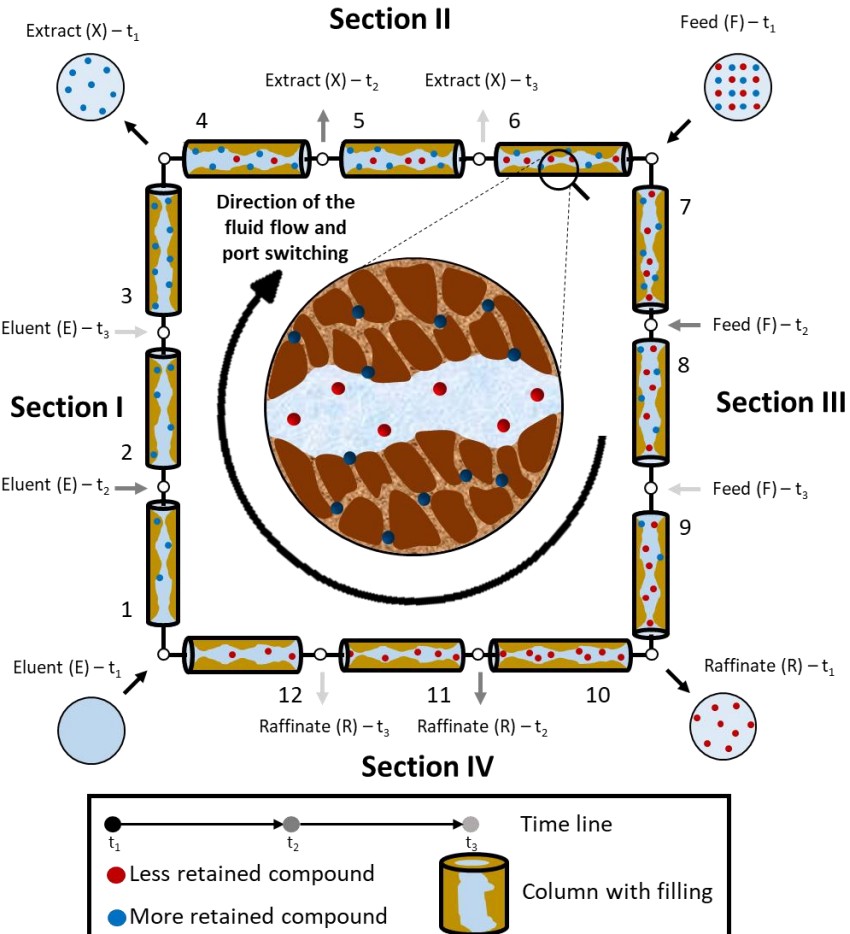

**Figure 14.** Schematic diagram of an SMB chromatograph and its operation where the more retained compound is adsorbed onto the solid porous phase.

Optimal SMB operation depends upon several factors, such as reasonable operating conditions of its streams, switching time, eluent, and the chiral stationary phase. Engineering practices have shown that the scenario of SMB operating conditions is rather limited; a poor set of the pump flowrates may lead to contamination of the outlet streams, thus compromising the separation. The Simulated Moving Bed has been gaining prominence due to its advantageous features such as cleanliness, small size, security, and fast procedure when compared to other systems. Moreover, the method has presented great performance in the separation of enantiomeric drugs such as praziquantel [209], which is a drug frequently used to combat schistosomiasis.

## 6. Conclusions

Due to the importance of enantiomer in several fields, especially for pharmaceutical companies that strive to diminish drug side effects, this article approaches different features of enantiomers. The article defines what an enantiomer is, presents related nomenclature, optical activity, enantiomers actuation in human body, enantiomers in biomolecules, market and pharmaceutical demands, and enantioresolution methods such as membranes, crystallization, and chromatography, especially TMB and SMB. It also reports on the features of the most applied enantioresolution methods, main enantiomers, eluents, and chiral stationary phases, with an emphasis on the cyclic adsorption processes. Through the comprehension of the aspects related to enantiomers, the authors of this articles hope to raise the interest of incoming students of chemical engineering to enantioresolution, control, and optimization field.

**Author Contributions:** Conceptualization, I.B.R.N., K.V.P. and R.S.; writing—original draft preparation, I.B.R.N., K.V.P. and R.S.; writing—review and editing, I.B.R.N., K.V.P. and R.S.; supervision, I.B.R.N. and K.V.P. All authors have read and agreed to the published version of the manuscript.

**Funding:** This study was financed in part by the Coordenação de Aperfeiçoamento de Pessoal de Nível Superior-Brasil (CAPES)-Finance Code 001. This work was also financially supported by: Base Funding-UIDB/50020/2020 of the Associate Laboratory LSRE-LCM-funded by national funds through FCT/MCTES (PIDDAC); FCT—Fundação para a Ciência e Tecnologia under CEEC Institutional program.

**Conflicts of Interest:** The authors declare no conflict of interest.

**Entry Link on the Encyclopedia Platform:** https://encyclopedia.pub/19483.

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
