# Peer review of "Enantiomers and Their Resolution"

_encyclopedia, doi:10.3390/encyclopedia2010011_

Round 1

Reviewer 1 Report

Dear Authors,

The subject 'enantiomers and their resolution' is important, especially in medicinal chemistry and natural active product. The manuscript is informative and useful. However, I have some concerns:

The abstract seems to be ‘too basic‘. It is a general knowledge that is present in any related typical textbook. Please rewrite the abstract to make sure that the abstract is the brief summary of the manuscript.

Figures. There is too much redundancy regarding the figures,for instance, Fig. 16, 17. Each figure only need to appear once. Please delete the redundant pictures. (fig. 9, fig. 15, 16, 17, 18, the pictures have appeared more than one time in the manuscript. In my opinion, these redundant pictures must be deleted ).

In addition, please make sure that the authors have accquired the using permissions of third party materials.

Some figures (like figures 1, 2, 3) are correct, but they are somewhat 'too basic'. I am not sure if these are necessary in an scientific paper.

Thank you!

Author Response

Thank you for your comments. Please see attached file.

Reviewer 2 Report

The topic of the hirality of organic compounds is a fundamental issue and at the same time very important because it affects their properties and practical application. The manuscript collects at the academic level basic data on stereochemistry in one place. Therefore, I suggest that it would be better if the drug structures presented in Tables 2 and 3 were presented graphically (their enantiomeric structures) and not only in name. Structures will be more informative than names. The separation of enantiomers is one of the important issues of preparative chemistry and the pharmaceutical industry, especially. Therefore, it is good that the topic of isolation and purification has been covered in the manuscript. 

Author Response

(The authors gave the same response as above.)

Round 2

Reviewer 1 Report

Dear Authors,

Thank you for your response.

The important issues have been addressed. The article is comprehensive, useful and informative.

There are now just some minor issues.

1) Please check the numbering of the Tables. Table 1, 2, ....

2) Figures 13 seems to occur twice (Page 25, 27), and so does Figure 14 (Page 30, 31). I wonder if this is necessary. 

3) There are many 'Error! Reference source not found' (Line 25, for example). Please check and confirms the citations.

In my opinion, as soon as these are addressed, the manuscript can be accepted for publication.

Thank you!

Author Response

Thanks for the comments, we appreciate the support. Please see attached file. 

Reviewer 2 Report

The manuscript looks better, although it could be more graphically polished. One of the figures (structures) are small, the other are large, or, as in table 2, there are structures overlapping the table border. 

Author Response

Thanks for the comments, we appreciate the support. We believe that there was a formatting problem when converting Word to PDF. We have reviewed the pictures and we believe now they have an appropriate size.